# The Influence of Carbides on Atomic-Scale Mechanical Properties of Carbon Steel: A Molecular Dynamics Study

**DOI:** 10.3390/nano12234179

**Published:** 2022-11-24

**Authors:** Liang Zhang, Longlong Yang, Kun Sun, Pujie Zhu, Keru Chen

**Affiliations:** State Key Laboratory for Mechanical Behavior of Materials, Xi’an Jiaotong University, Xi’an 710049, China

**Keywords:** MD simulation, ferrite-carbide structure, carbide type, mechanical properties

## Abstract

Pearlite is an important structure in carbon steel; however, the influence mechanism of carbides in pearlite on its mechanical properties and microstructure evolution has not yet been fully elucidated. In this work, a ferrite–carbide composite model with various carbide types was constructed to investigate the influence of carbide types via a uniaxial compression deformation using classical molecular dynamics simulations. It was found that the carbide type had little effect on the compressive elastic modulus, but a more obvious effect on the yield strain, yield stress, and flow stress. The maximum compressive elastic modulus was in the Fe2C model, with 300.32 GPa, while the minimum was found in the Fe4C model at 285.16 GPa; the error was 5.32%. There were significant differences in the yield stress, yield strain, and flow stress of the ferrite–carbide model according to the stress–strain curve. Secondly, the type of carbide used affected its elastic constant, especially the bulk modulus and Cauchy pressure. The maximum bulk modulus of the Fe4C model was 199.01 GPa, the minimum value of the Fe3C model was 146.03 GPa, and the difference was 52.98 GPa. The Cauchy pressure calculation results were consistent with the yield strain trend. Additionally, the effective elastic moduli of the composite system were used to verify the accuracy of the calculation results of this work. Thirdly, ferrite–carbide interfaces could act as a resource for dislocation emission. The initial stacking fault forms at ferrite–carbide interfaces and expands into ferrite. The dislocation type and segment in the ferrite–carbide model were significantly different due to the type of carbide used.

## 1. Introduction

Carbon steel is the most widely used metal material in the machinery manufacturing industry, owing to its mechanical and processing properties [1,2]. The microstructures of carbon steel have been extensively explored for decades. It is well known that pearlite composed of the lamellae of cementite (Fe3C, orthorhombic structure) and ferrite (α-Fe, body-centered cubic, BCC) is a common microstructure in carbon steel [3]. The synergistic effects between α-Fe and Fe3C phases afford pearlite a high strength [4,5], and these synergistic effects include interface strengthening between α-Fe and Fe3C phases, solid solution strengthening through cementite decomposition [6,7,8], and dislocation strengthening [9,10]. Thus, understanding the behavior of α-Fe and carbide is fundamental to studying the mechanical response of carbon steel.

Different morphologies and types of carbide and ferrite form various structures, such as pearlite and bainite. In recent years, extensive studies on the microstructure of carbon steel have been carried out to explain its unique mechanical properties. Backscattered electron imaging (BSE) and electron backscatter diffraction (EBSD) have been employed to characterize aspects of the microstructure of pearlite, for instance, the pearlite nodule, interlamellar spacing, morphology of pearlite, and the orientation of α-Fe [11]. Guo et al. also discussed the pearlite colony and the low-angle boundaries in the α-Fe matrix. Meanwhile, researchers have revealed the nucleation and growth mechanism of pearlite. Varshney et al. [12] investigated nanocarbides in high silicon steels and analyzed the formation mechanism of nanocarbides. Furthermore, the relationship between microstructure parameters (pearlite colony size and austenite grain size) and mechanical properties (strength and impact toughness) was studied by Zhang [13]. 

At the nanoscale, the experimental cost is high to study the effect of carbides on the mechanical properties of carbon steel. Fortunately, molecular dynamics (MD) simulations coincidentally make up for the lack of experimental exploration of material properties at the nanoscale. MD simulations have been widely used to investigate the properties of pearlite. Guizewski et al. [3] constructed an atomistic model with the various orientation relationships between α-Fe and Fe3C and investigated the interfacial effects on the pearlite microstructure. Shimokawa et al. [14] studied the effect of interfacial dislocation at the phase interface on the ductility of drawn pearlite. In addition, the properties of Fe3C in pearlite were studied [15,16,17]. These research results show that computational simulations have been widely used in the research of carbides in carbon steel.

However, the effect of carbides types on the mechanical properties of carbon steel are less frequently discussed. Thus, in this work, a multilayer ferrite–carbide structure consisting of lamellar ferrite and carbide alternately was constructed. Uniaxial compression was then performed to investigate the effect of carbide type on the multilayer ferrite–carbide structure’s properties using MD simulations. The type of carbide in the ferrite–carbide structure was varied to explore the effects of interface and carbide types on microstructure evolution and its mechanical properties.

## 2. Modeling and Simulation Methodology

MD simulations were performed using the large-scale atomic/molecular massively parallel simulator (LAMMPS) package [18,19]. As shown in Figure 1, in the *y* direction, the initial model consisted of ferrite and various types of carbide “sandwiched” together. In the *z* direction, the initial model was divided into three zones: fixed layers, Newtonian regions, and loading layers. The total length along the *x* and *z* axis was 151.315 Å and 199.85 Å, respectively. The thickness ratio of ferrite and carbide on the *y* axis was approximately 1:1. To ensure a perfect lattice of carbides on the *y* axis, the thickness of carbides on the *y* axis was slightly different. The total length along the *y* axis was 158.746–166.285 Å. There were 454,817–502,120 atoms in the multilayer ferrite–carbide system. For convenience during analysis, the models were named after the carbide formula that constituted the model, namely, the Fe2C model, Fe3C model, Fe4C model, and Fe7C3 model.

The calculated lattice constant of ferrite was 2.855 Å, which was in agreement with the experiment value of a = 2.87 Å [20]. In this work, as shown in Figure 1, there were four models containing different types of carbide. The detailed parameters of the carbides are shown in Table 1. The calculated parameters of the different carbides were in good agreement with the experiment values [16,21,22,23].

The commonly reported orientation relationships (ORs) for ferrite and carbide are the Bagaryatskii OR, Pitsch–Petch OR, and the Isaichev OR, and their associated orientations [24,25,26]. The specific crystallographic orientations for the various ORs are shown in Table 2. Among the various ORs, the Bagaryatskii OR strongly depends on the thickness and volume ratios of carbide [3,26]. Hence, in this work, the coordinate systems of all the multilayer ferrite–carbide models along the *x*-, *y*-, and *z*-axis were determined with the Bagaryatskii OR. As shown in Figure 1, the specific crystallographic orientation along the *x*-, *y*-, and *z*-axis were [$\bar{1}$10]α || [100]θ, [111]α || [010]θ, and (11$\bar{2}$)α || (001)θ, where α and θ indicate the ferrite and carbide phases, respectively.

In the present work, the modified embedded atom model (MEAM) potential developed by Mori et al. [27] was used to depict the atomic interaction in ferrite, and the Tersoff potential developed by Henriksson et al. [28] was used to depict the atomic interaction in carbide and the interaction of atoms at the ferrite and carbide interface. As with the original embedded atom model (EAM) potential, the modified embedded atom method (MEAM) is an embedding energy term, and the MEAM potential is an extension of the EAM potential. In the formulation of the MEAM, the total energy E can be obtained as
(1)$$E=\sum_{i}\left\{F_{i}\left(\overset{-}{\rho_{i}}\right)+\frac{1}{2}\sum_{i\neq j}\varphi_{ij}\left(r_{ij}\right)\right\}$$
where F is the embedding energy, ρ and φ are a pair potential interaction, and i and j are the atoms within the cutoff distance. In the present work, this MEAM potential developed by Mori et al. clarified the dislocation dynamics in BCC iron based on density functional theory (DFT) calculations.

The Tersoff potential is a three-body potential, and the energy E is described as
(2)$$E=\frac{1}{2}\sum_{i}\sum_{i\neq j}f_{C}\left(r_{ij}\right)\left[f_{R}\left(r_{ij}\right)+b_{ij}f_{A}\left(r_{ij}\right)\right]$$
(3)$$f_{C}\left(r\right)=\left\{\begin{array}{l} 1:r<R-D\\ \frac{1}{2}-\frac{1}{2}\sin\left(\frac{\pi }{2}\frac{r-R}{D}\right):R-D<r<R+D\\ 0:r>R+D \end{array}\right.$$
where $f_{R}$ and $f_{A}$ are the two-body and three-body interactions, respectively. The energy of atom i is all neighbors j and k within a cutoff distance equal to *R + D*. This Tersoff potential developed by Henriksson et al. reproduces the lattice parameters and formation energies of the principal ferrite and carbide, with excellent agreement with empirical results. Thus, in the present work, the MEAM and Tersoff potentials were used to compute pairwise interactions for atoms in the multilayer ferrite–carbide system. 

The *x* and *y* orientation were set up with a periodic boundary, and the *z* orientation was set up with a nonperiodic boundary that it was shrink-wrapped with a minimum value boundary. At the simulation’s initial stage, the composite structure was relaxed using energy minimization to adjust all the atoms’ coordinates. Then, the simulated structure’s temperature was set to 300 K and equilibrated thermally at 300 K for 150 ps by imposing a Nose–Hoover thermostat isothermal-isobaric (NPT) ensemble to release the stress and obtain a stable ferrite–carbide composite system. For the uniaxial compress loading, the fixed layers and loading layers were set as a rigid system. The loading layers moved at a displacement speed of 1.8 m/s, which was calculated using a strain rate of 10^8^/s. The direction of displacement was along the Z direction, as shown in Figure 1. Common neighbor analysis (CNA) [29] was employed to characterize the local structure, and the dislocation extraction algorithm (DXA) [30,31] was employed to identify dislocation defects. The analysis method used in this work was similar to previous studies [26]. The open visualization tool (OVITO) [32] was used to visualize the atomic configurations and calculate the atomic strain.

## 3. Results and Discussions

### 3.1. Stress-Strain Curves of the Ferrite-Carbide Systems

During the uniaxial compression deformation, the stress and strain of the multilayer ferrite–carbide composite systems were monitored and plotted on typical stress–strain curves. Figure 2 presents the results of the uniaxial compression simulation applied to the multilayer ferrite–carbide composite systems with various types of carbide. On the whole, all stress–strain curves of the ferrite–carbide systems were divided into two fundamental stages: the elastic stage and the plastic stage. At the initial deformation stage, the stress linearly increased with increasing strain, as shown in Figure 2. The stress increased up to a stress threshold, which corresponded to the yield point in the compression test. Then, the stress reduced slightly and the ferrite–carbide models showed that dislocations were generated, and that extended dislocations and dislocation multiplication occurred. As shown in Figure 2, the type of carbide had an insignificant effect on Young’s modulus values but had a significant effect on the yield stress and yield strain. After the ferrite–carbide models’ yield was accounted, the flow stress of models was still significantly different. The strain was 7.61–10.86% and the flow stress of the Fe2C model was significantly greater than that of other models. Why this happened and how the carbide lamella affected the mechanical properties will be analyzed in the next section.

By fitting the slope of the stress–strain curve of the ferrite–carbide models in the elastic stage, the compressive elastic modulus of each model was obtained, as shown in Figure 3. For all ferrite–carbide composite models, the type of carbide had an insignificant effect on compressive elastic modulus values. The highest compressive elastic modulus value was 300.32 GPa and was obtained for the Fe2C model, while the lowest compressive elastic modulus value was 285.16 GPa, obtained for the Fe3C model. The maximum difference in compressive elastic modulus was 5.32%. The bond energy was the main factor affecting the Young’s modulus values, with a greater bond energy resulting in a greater Young’s modulus value [33]. Thus, based on the total energy of the ferrite–carbide composite systems, the mean bond energy of each ferrite–carbide composite model was calculated. The results showed that the mean bond energy of the Fe2C model, Fe3C model, Fe4C model and Fe7C3 model was −5.134 eV/atom, −4.822 eV/atom, −5.099 eV/atom, and −4.896 eV/atom, respevtively. In general, the lower the bond energy was, the more stable the specimen was. There was a good correspondence between the bond energy and compressive elastic modulus.

Through the stress–strain curve of the ferrite–carbide models, it was found that the type of carbide had a significant effect on the yield stress and yield strain, the yield strain and stress of the Fe2C model in particular were greater than those of other models. As shown in Figure 4, the yield strain was 8.31%, 7.39%, 7.02%, and 7.63% for the different models. On one hand, the compressive yield stress and yield strain were related to the properties of the carbide itself. The carbon content in each ferrite–carbide model was counted, with 13.8%, 6.69%, 9.8%, and 11.8% in each model. Interstitial carbon atoms increased the yield stress and strain of the ferrite–carbide models. On the other hand, from previous research results, it is known that the characteristics of ferrite are that it is soft and tough, while carbide is brittle and hard [34]. Therefore, yielding preferentially occurs inside carbide or at ferrite–carbide interfaces (FCIs). Kim et al. [24] studied the effect of misfit dislocations at FCIs on the mechanical properties and phase transformation of pearlitic steel. In the present work, the FCIs composed of ferrite and different types of carbide affected the yield strain and stress of the ferrite–carbide models. The introduction of an interface in the ferrite–carbide model led to an increase in the energy of the system compared to the energy in pure ferrite after relaxation, so the energy change due to FCIs was calculated. In the present work, the average energy of atoms in the ferrite and carbide was employed to calculate the theoretical energy of the ferrite–carbide composite system. The ΔE was estimated using the difference between the theoretical and actual energy. The ΔE of the ferrite–carbide model was 0.0497 eV/atom, 0.3689 eV/atom, 0.5476 eV/atom, and 0.1970 eV/atom for the models, as shown in Table 3. It is well known that the higher the energy, the more unstable the system. Due to the FCIs, the Fe4C model had the greatest ΔE increase, so its yield strain was the smallest. Similarly, the Fe2C model had the smallest ΔE and the biggest yield strain.

As shown in Figure 4, the yield stress of the ferrite–carbide models was 25.42 GPa, 21.43 GPa, 20.35 GPa, and 21.56 GPa. In addition to the above-mentioned influence of FCIs, the von Mises stress distribution was analyzed and is shown in Figure 5. The von Mises stress in the Fe2C model was the smallest and was mainly distributed on the FCIs, while the von Mises stress in the other models was greater and was mainly distributed in the carbide. The greater the von Mises stress was at the interface, the lower the von Mises stress required for dislocation nucleation was, which illustrated the yield strain and stress corresponding to Figure 4.

### 3.2. Elastic Constants of the Ferrite-Carbide Systems

The stress–strain curves of the ferrite–carbide models and the effect of the carbide type on yield strength and Young’s modulus were analyzed in Section 3.1. In this section, the effect of the carbide type on the elastic constants is analyzed. Elastic constants are key mechanical parameters of materials. Previously, researchers have attempted to calculate elastic constants of materials based on density functional theory (DFT). In the present work, MD simulations were employed to calculate the elastic constants of the ferrite and ferrite–carbide models. The calculated conditions were the periodic boundary conditions, with the temperature at 300 K, and the deformation of the models was set to 2%. The calculated results are shown in Table 4.

According to the MD-calculated principle elastic constant values (C11, C22, C33, C12, C13, C23, C44, C55, and C66) in Table 4, the bulk modulus (*B*), shear modulus (*G*), Poisson’s ratio (*γ*), Young’s modulus (*E*), and Cauchy pressure were also calculated. The calculated results are shown in Table 5. The bulk modulus *B* of a crystal was determined using the resistance of chemical bonds to compression and it reflects the macroscopic properties of materials. The bulk modulus *B* of the ferrite–carbide models was 199.01 GPa, 177.85 GPa, 146.03 GPa, and 182.21 GPa. This bulk modulus trend was consistent with the compressive elastic modulus. Due to the various Δ energy and FCIs in the ferrite–carbide models, as described in the previous section, the type of carbide had a certain influence on the bulk modulus *B* of the ferrite–carbide models. The shear modulus *G* characterizes the ability of a material to resist shear strain, indicating how easily the material deforms under shear strain. According to the calculated shear modulus values in Table 5, the Fe2C model had the highest elastic modulus value, while the Fe4C model had the lowest elastic modulus value. Poisson’s ratio *γ* is the ratio of the transverse strain to the longitudinal strain of the model and it reflects the elastic constant of the lateral deformation of the material. In a study by Takaki et al. [35], the Poisson’s ratio *γ* of ferritic and martensitic steels was identical at 0.31. The Poisson’s ratio *γ* of the ferrite–carbide models was approximately 0.31, which was close to the results of previous studies [36]. Due to the different characteristics of various types of carbide, the Poisson’s ratio *γ* also exhibited certain differences, as shown in Table 5. Young’s modulus *E* describes the ability of a solid material to resist deformation and measures the stiffness of an isotropic elastomer. According to the calculated Young’s modulus values in Table 5, the Fe2C model had the highest Young’s modulus value of 215.49 GPa, while the Fe4C model had the lowest Young’s modulus value of 173.53 GPa. If the Cauchy pressure calculated from C12 to C44 was greater than zero, the alloy was ductile, and the greater the values, the better the plastic behavior of the alloy. In Table 5, the highest Cauchy pressure of 62.92 GPa was in the Fe2C model, and the lowest value was 34.96 GPa, from the Fe4C model, which is consistent with the variation trend in the yield strain of various ferrite–carbide models shown in Figure 4.

The ferrite–carbide model is the lamellar structure of ferrite and carbide, so it is possible to make an approximation of its mechanical response using a composite [3,26]. The effective elastic moduli of the composite system can be described as
(4)$$E_{effective}=\frac{E_{\alpha }V_{\alpha }}{V}+\frac{E_{\theta }V_{\theta }}{V}$$
where E and V are the elastic modulus and volume of the individual lamellar, respectively. This formulation shows that when the ratio of ferrite and carbide is constant, the effective elastic modulus is also constant, regardless of the thickness of the carbide. Therefore, the Young’s modulus of a single crystal of ferrite and carbide was calculated to verify the reliability of the calculated results and the results are shown in Figure 6. From Figure 6, it can be seen that the Young’s moduli obtained by the fitting of the three methods were relatively similar, and the largest error was that the Young’s modulus of the Fe4C model was 7.34%. This result showed the reliability of the calculated results in this work. The Young’s modulus results show that the type of carbide was the main factor determining the elastic constant of the ferrite–carbide model when the thickness of the carbide was constant. This was consistent with the results of the analysis in the previous section.

In this section, elastic constants (the bulk modulus, shear modulus, Poisson’s ratio, Young’s modulus, and Cauchy pressure) for different ferrite–carbide models were calculated, and the Young’s modulus values were verified. The results showed that the changing trend of the elastic constant was consistent with the trend of the compressive elastic modulus shown in Section 3.1. Although the compressive elastic modulus and elastic constant of the Fe2C model were both the maximum values obtained, the most common type of carbide in carbon steel is Fe3C. To explain this phenomenon, in addition to the basic properties of carbides shown in Table 1, the formation energy and densities of various carbide types were analyzed and are shown in Table 6 [37,38]. The formation energy refers to the energy released by the corresponding elemental synthetic compound. When judging the stability of a material with energy, it may be more practical to select the formation energy [39]. Carbides exist stably in the organization, and not only the formation energy, but also parameters such as density must be considered. Carbides of the Fe3C type have the best combination of formation energy and density. Therefore, in a carbon steel structure, the most common carbide used is Fe3C. In the next section, an analysis of the evolution of various types of carbide during compressive deformation is shown.

### 3.3. Effect of Carbide Type on the Microstructural Evolution

To better understand the type of carbide in the microstructural evolution, the evolution of dislocation nucleation was first monitored, and some significant nucleation configurations were carefully investigated. To facilitate the observation of defects inside the model, the atomic configuration of the model under a certain strain after the yield point was selected to monitor the model, and the results are shown in Figure 7. For the Fe2C and Fe3C models, the initial dislocations were formed in the upper zone of the model, while the initial dislocations of the Fe4C and Fe7C3 models were in the central zone. Although the initial dislocation of the ferrite–carbide models was on a different side to the FCIs, given the model symmetry, it is not discussed in this work. From Figure 7, it is worth noting that, although dislocation embryos were formed on the FCIs, the location of the dislocation embryos was significantly different. The initial dislocations formed on the FCIs and then expanded into the ferrite. It can be noted that the dislocation nucleation formed not only on the FCIs, but also in the ferrite, as shown in Figure 7d.

To further understand the effects of carbides in the plastic stage, the atomic structures of the second stress peak in the stress–strain curve are shown in Figure 8. The strain at the second stress peak of the ferrite–carbide models was 9.69%, 8.79%, 8.36% and 8.84%. According to the atomic configuration, it can be seen that, for the Fe2C model, there were two dislocation types 1/2 <111> and <100>, 15 dislocation segments, and the total length of dislocations was 848.915 Å. For the Fe3C model, there were two dislocation types 1/2 <111> and <110>, 8 dislocation segments, and the total length of dislocation was 788.878 Å. For the Fe4C model, there was only one dislocation type 1/2 <111>, 4 dislocation segments, and the total length of dislocation was 781.799 Å. For the Fe7C3 model, there were two dislocation types 1/2 <111> and <100>, 18 dislocation segments, and the total length of dislocation was 1224.801 Å. From Figure 8b, it can be seen that there were more dislocation embryos in the Fe3C model compared with the other models. Therefore, based on the second stress peak in the stress–strain curve shown in Figure 2, the Fe3C model had the greatest strain and stress increase. It can be seen that there was a lot of dislocation at the interface between the fixed layers and the Newtonian region. This was due to the two regions being set up using different ensembles and is not discussed in this work.

The radial distribution function (RDF) is a density–density correlation function. The RDF is described as follows [40]:(5)$$RDF=g\left(r\right)=\frac{1}{4\pi r^{2}N\rho }\sum_{i=1}^{N}\sum_{\begin{array}{l} j=1\\ j\neq i \end{array}}^{N}\delta \left(r-r_{ij}\right)$$
where *N* is the total number of atoms (in this work, *N* = 200), *ρ* is the density of the system, *δ* is the Kronecker constant, and $r_{ij}$ is the distance between atoms *i* and *j*. In the past, the RDF has been used to indicate structural transformations during contact loading [15,41]. Thus, the RDF was used to monitor the interatomic bond length between Fe and C atoms of carbides in this work. The results of carbides during relaxation, at the yield point and at the second stress peak were compared in the graphs shown in Figure 9. All carbides exhibited two obvious crystalline peaks (A and B), as shown in Figure 9, and the results of the interatomic bond length at the two peaks are shown in Table 7. It can be seen from Table 7 and Figure 9a that, before compression, Fe2C exhibited a crystalline peak at an interatomic bond length of 1.7385 Å, and a corresponding second peak at 2.455 Å. At the yield point and second stress peak, the first peak (A) did not show even a slight change. However, the second peak (B) showed a noticeable change in the count of the number of atoms. It is worth noting that, unlike other carbides, the count of the number of atoms at B decreased for Fe2C, while the count of Fe3C, Fe4C, and Fe7C3 increased, as shown in Figure 9. Compared to Figure 9, the pattern of the high order peaks of Fe2C and Fe4C became disordered, but that of Fe3C and Fe7C3 remained relatively orderly. This phenomenon was related to the density of carbides, as shown in Table 6, and the density of Fe3C and Fe7C3 was greater than that of Fe2C and Fe4C.

## 4. Conclusions

In summary, multilayer ferrite–carbide models were placed under uniaxial compression to explore the effects of ferrite–carbide interfaces and the carbide type on the mechanical properties and microstructure evolution using a molecular dynamics simulation.

Firstly, the stress–strain curves of the ferrite–carbide models under uniaxial compression could be divided into the elastic stage and the plastic stage. In the elastic stage, the type of carbide had an insignificant effect on compressive elastic modulus values. The maximum compressive elastic modulus was obtained for the Fe2C model and was 300.32 GPa, while the minimum was that of the Fe4C model at 285.16 GPa. However, the type of carbide had a significant effect on the yield stress and yield strain, with the yield strain and stress of the Fe4C model in particular being lower than the other models. In the plastic stage, the flow stress of the Fe2C model was greater than the other models.

Secondly, the effect of the carbide type on the elastic constants of the ferrite–carbide models was investigated. The carbide type had a certain influence on the bulk modulus and Cauchy pressure, while its influence on the shear modulus, Young’s modulus and Poisson’s ratio was not obvious. The elastic modulus calculation results were also verified by the effective elastic moduli of the composite system theory.

Thirdly, dislocation embryos were formed at different locations of FCIs. The atomic configuration at the second stress peak was analyzed. It was found that the dislocation type and segment in the ferrite–carbide model were significantly different due to the type of carbide used.

## Figures and Tables

**Figure 1 nanomaterials-12-04179-f001:**
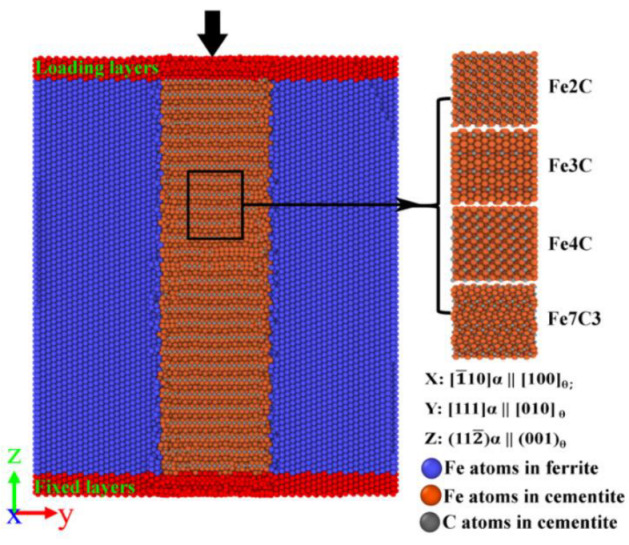
Initial model of the ferrite–carbide structure with various types of carbide.

**Figure 2 nanomaterials-12-04179-f002:**
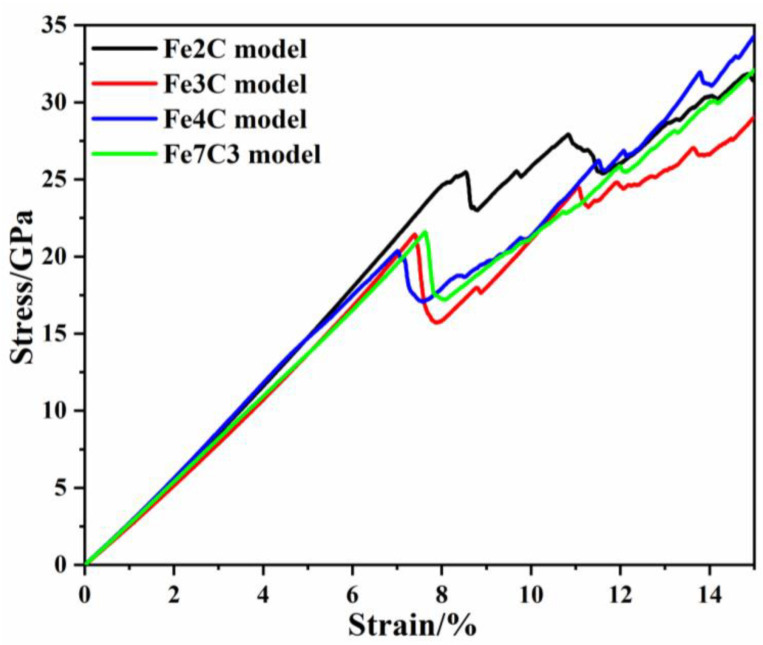
Stress–strain curve of all ferrite–carbide models.

**Figure 3 nanomaterials-12-04179-f003:**
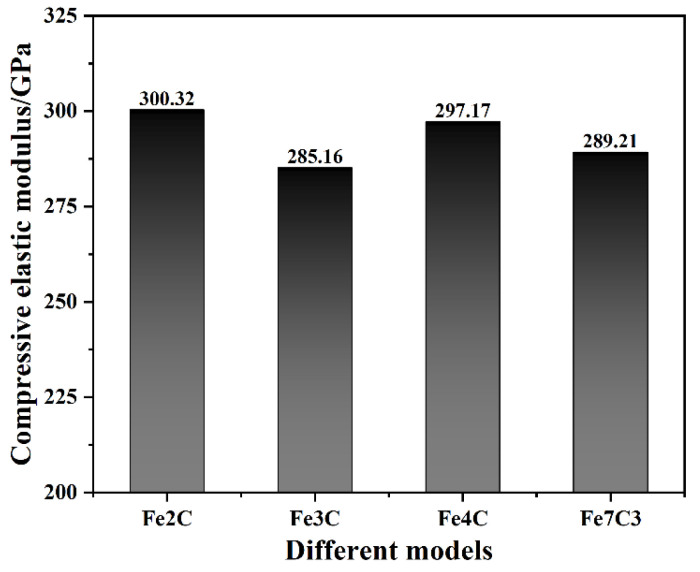
Compressive elastic modulus of all ferrite–carbide models.

**Figure 4 nanomaterials-12-04179-f004:**
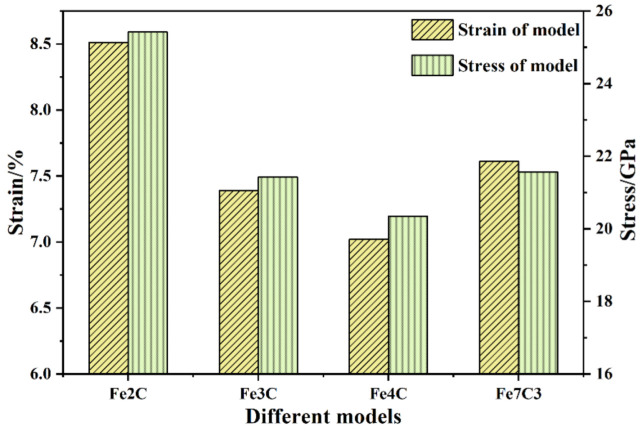
Strain and stress of all ferrite–carbide models.

**Figure 5 nanomaterials-12-04179-f005:**

**Von-Mises** stress distribution in the (**a**) Fe2C model; (**b**) Fe3C model; (**c**) Fe4C model; (**d**) Fe7C3 model.

**Figure 6 nanomaterials-12-04179-f006:**
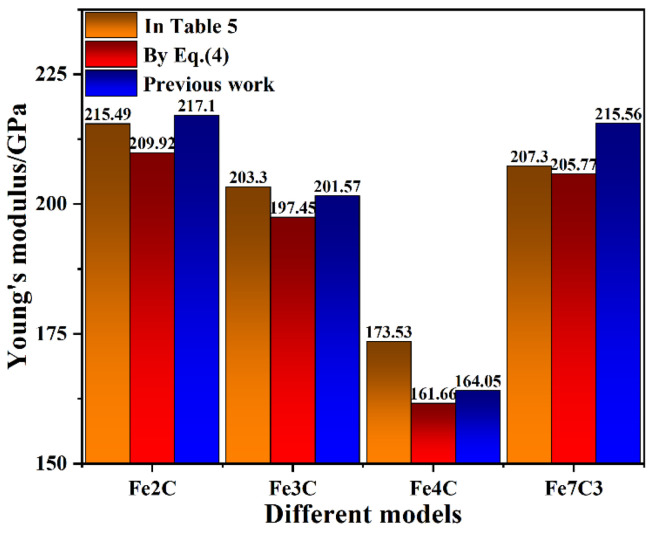
The fitted Young’s modulus values of the ferrite–carbide composite model.

**Figure 7 nanomaterials-12-04179-f007:**
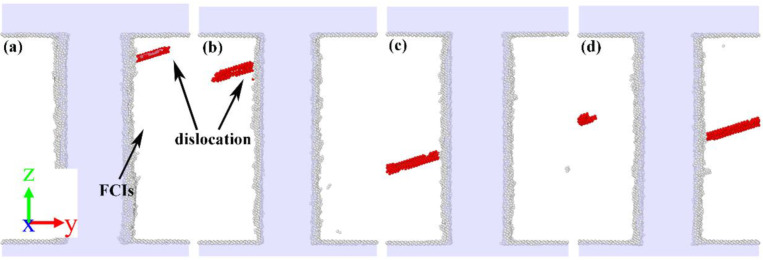
Atomic configurations for the dislocation nucleation of (**a**) Fe2C model, (**b**) Fe3C model, (**c**) Fe4C model, and (**d**) Fe7C3 model. The red atoms are the stacking fault atoms, and the white atoms are the disordered atoms at FCIs and boundary.

**Figure 8 nanomaterials-12-04179-f008:**
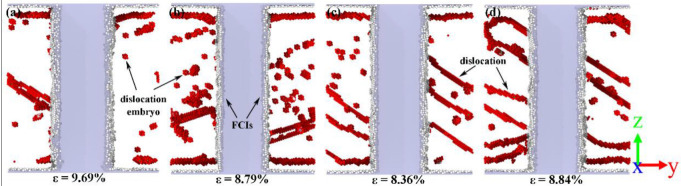
The atomic structure at the second stress peak. (**a**) Fe2C model, (**b**) Fe3C model, (**c**) Fe4C model, and (**d**) Fe7C3 model. The red atoms represent the dislocation atoms and the gray atoms represent the FCIs atoms.

**Figure 9 nanomaterials-12-04179-f009:**
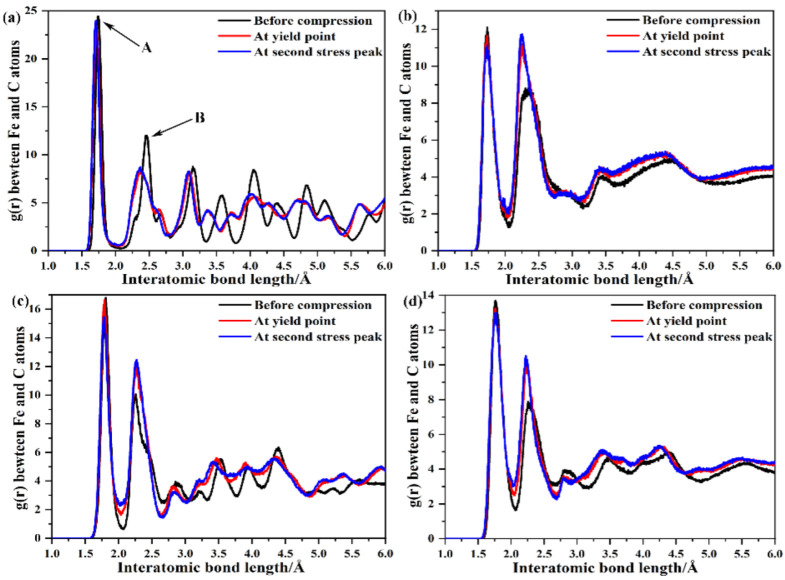
Radial distribution function between iron and carbon atoms (**a**) Fe2C, (**b**) Fe3C, (**c**) Fe4C, (**d**) Fe7C3.

**Table 1 nanomaterials-12-04179-t001:** Calculated parameters of carbide using MD.

Formula	Space Group	Lattice/Å	Crystal System	Nsites	Volume/Å^3^
Fe2C	Pnnm	a = 2.824, b = 4.282, c = 4.714	orthorhombic	6	56.982
Fe3C	Pnma	a = 4.491, b = 5.03, c = 6.739	orthorhombic	16	152.239
Fe4C	P-43m	a = b = c = 3.842	cubic	5	56.692
Fe7C3	Pnma	a = 4.517, b = 6.857, c = 11.762	orthorhombic	40	364.342

**Table 2 nanomaterials-12-04179-t002:** Orientation of the various ORs.

Parameter	Orientation X	Orientation Y	Orientation Z
Bagaryatskii OR	[$\bar{1}$10]α || [100]θ	[111]α || [010]θ	(11$\bar{2}$)α || (001)θ
Near Bagaryatskii OR	$[\bar{1}\;12\bar{\;11}$]α || [100]θ	[111]α || [010]θ	($23\;\bar{10}\;13$)α || (001)θ
Pitsch-Petch OR	$[\bar{10}\;31\bar{\;12}$]α || [100]θ	$[\bar{11}\;10\;35$]α || [010]θ	[521]α || (001)θ
Near Pitsch-Petch OR	$[\bar{12}\;39\bar{\;17}$]α || [100]θ	[$\bar{1}$13]α || [010]θ	[134 53 27]α || (001)θ
Isaichev OR	$[1\bar{1}0$]α || [100]θ	[111]α || [010]θ	$[11\bar{2}$]α || (001)θ

**Table 3 nanomaterials-12-04179-t003:** MD-calculated Δ energy of various ferrite–cementite models.

Model	Fe2C Model	Fe3C Model	Fe4C Model	Fe7C3 Model
ΔE/(ev/atom)	0.0497	0.3689	0.5476	0.1970

**Table 4 nanomaterials-12-04179-t004:** MD-calculated elastic constants of the ferrite–carbide models (unit: GPa).

Constants	Fe2C Model	Fe3C Model	Fe4C Model	Fe7C3 Model
C11	373.95	311.43	250.99	306.65
C22	347.23	292.31	211.72	286.32
C33	290.61	305.73	254.79	302.82
C12	111.54	111.06	93.55	119.99
C13	69.51	103.16	93.83	110.05
C23	72.48	101.74	94.41	119.82
C44	48.62	62.59	58.59	69.61
C55	71.36	60.15	47.54	59.27
C66	83.04	78.53	56.88	78.51

**Table 5 nanomaterials-12-04179-t005:** MD-calculated elastic properties of the ferrite–carbide models.

Constants	Fe2C Model	Fe3C Model	Fe4C Model	Fe7C3 Model
*B*/GPa	199.01	177.85	146.03	182.21
*G*/GPa	81.654	77.63	66.642	79.098
*γ*	0.3195	0.3095	0.3019	0.3104
*E*/GPa	215.49	203.30	173.53	207.30
Cauchy pressure/GPa	62.92	48.47	34.96	50.38

**Table 6 nanomaterials-12-04179-t006:** Formation energy and density of various carbides.

Parameter	Fe2C	Fe3C	Fe4C	Fe7C3
Formation energy/(ev/atom)	0.226	0.055	0.514	0.178
Density/(g/cm3)	7.21	7.83	6.89	7.78

**Table 7 nanomaterials-12-04179-t007:** Output results of interatomic bond length at two peaks.

*g*(*r*)/Å	Before Compression		Yield Point		Second Stress Peak	
	A Point	B Point	A Point	B Point	A Point	B Point
Fe2C	1.738	2.455	1.720	2.371	1.705	2.362
Fe3C	1.729	2.353	1.729	2.248	1.729	2.248
Fe4C	1.804	2.257	1.789	2.266	1.785	2.266
Fe7C3	1.759	2.254	1.756	2.248	1.762	2.224

## Data Availability

The authors declare that all data supporting the finding of this study are available within the article.

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
