# Peer review of "The Influence of Carbides on Atomic-Scale Mechanical Properties of Carbon Steel: A Molecular Dynamics Study"

_nanomaterials, 2022, doi:10.3390/nano12234179_

Round 1

Reviewer 1 Report

Dear authors, you can find my report in the attached file.

Kind regards!

Reviewer 2 Report

The article is devoted to the study by the method of molecular dynamics of the mechanical properties of a composite consisting of a carbide (Fe2C, Fe3C, Fe4C or Fe7C3) strip in a ferrite matrix. The task is interesting and non-trivial. The used method, software, potentials are modern. My remarks are the following:

1) It is necessary to explain in the article why such a model was chosen: a strip of one or another carbide in a ferrite matrix? Why a strip and not a particle in a ferrite? Why is only compression directed strictly along the carbide strip considered? These explanations are not in the article.

2) A serious remark concerns the boundary conditions in the model. When modeling deformation, especially plastic deformation, it is necessary to carefully approach this and take into account the transverse expansion of the material during compression or, when dislocations appear, their free movement, not limited by boundary conditions. However, periodic boundary conditions are imposed on the lateral faces of the simulated parallelepiped, which, by definition, fix the dimensions of the computational cell along the X and Y axes and prevent the transverse expansion of the material during compression and the free exit of dislocations during plastic deformation. The model used does not look physically adequate, and the results obtained lose their value. It is necessary to clarify this point in the article.

3) In the description of the procedure for creating a computational cell, nothing is said about taking into account the repeat periods (lattice parameters) of the crystal structures of ferrite and carbide. If periodic boundary conditions are used, then the dimensions of the computational cell along the X axis must be a multiple of the repetition periods of the ferrite and carbide structures, otherwise unnecessary internal stresses will be created.

4) What is the energy ΔE shown in Table 3? How exactly was it calculated? The text about this is said briefly and incomprehensibly, but in any case it is huge. This energy is enough to melt the material.

5) On Fig. 1 and in Table 1, the directions of the axes are indicated using different brackets: square and round. Round brackets usually denote planes. On Fig. 1, besides this, the minus in the indices is shown non-standard.

6) The second part of the paragraph before Table 1 consists of several identical sentences following one after another. It reads badly. It is better to rewrite it in different sentences or combine it into one.

7) Fig. 3 is completely unnecessary, the information is duplicated in the text.

8) Fig. 6 is also not needed - it duplicates part of the data from Table 5. It is better to expand the Table and not include an additional figure.

Round 2

Reviewer 1 Report

Dear authors, thank you so much for your kind reply and for taking into account my suggestions. Congratulations for your work. I believe that the work it is ready to be published after minor text editing (after updating the references to the mdpi style for example; but this it is another history that it implies to the mdpi-editors!)

King regards!
